# Learning about Victims of Holocaust in Virtual Reality: The Main, Mediating and Moderating Effects of Technology, Instructional Method, Flow, Presence, and Prior Knowledge

Miriam Mulders

Educational Media and Knowledge Management, University of Duisburg-Essen, Universitätsstraße 2, 45127 Essen, Germany; miriam.mulders@uni-duisburg-essen.de

**Abstract:** The goal of the current study was to investigate the effects of a virtual reality (VR) simulation of Anne Frank's hiding place on learning. In a 2 × 2 experiment, 132 middle school students learned about the living conditions of Anne Frank, a girl of Jewish heritage during the Second World War, through desktop VR (DVR) and head-mounted display VR (HMD-VR) (media conditions). Approximately half of each group engaged in an explorative vs. an expository learning approach (method condition). The exposition group received instructions on how to explore the hiding place stepwise, whereas the exploration group experienced it autonomously. Next to the main effects of media and methods, the mediating effects of the learning process variables of presence and flow and the moderating effects of contextual variables (e.g., prior technical knowledge) have been analyzed. The results revealed that the HMD-VR led to significantly improved evaluation, and—even if not statistically significant—perspective-taking in Anne, but less knowledge gain compared to DVR. Further results showed that adding instructions and segmentation within the exposition group led to significantly increased knowledge gain compared to the exploration group. For perspective-taking and evaluation, no differences were detected. A significant interaction between media and methods was not found. No moderating effects by contextual variables but mediating effects were observed: For example, the feeling of presence within VR can fully explain the relationships between media and learning. These results support the view that learning processes are crucial for learning in VR and that studies neglecting these learning processes may be confounded. Hence, the results pointed out that media comparison studies are limited because they do not consider the complex interaction structures of media, instructional methods, learning processes, and contextual variables.

**Keywords:** virtual reality; head-mounted displays; learning; Anne Frank; instructional design; media vs. methods; learning processes; flow; presence; prior knowledge; media comparison; complex interaction structures

## 1. Introduction

Virtual reality (VR) is one of several technologies receiving increasing attention in various educational settings. VR is expected to be used widely in classroom teaching within a few years [1], but its educational potential has not been thoroughly investigated because it has only recently seen widespread consumer availability.

VR is understood as a computer-mediated simulation that is three-dimensional (3D), multisensory, and interactive. The user is able to inhabit and act within an external environment [2]. VR enables unique learning scenarios, as simulations allow students to act as if they were in the real world while interacting with otherwise intangible or inaccessible objects [3,4]. VR provides students with the experience of a different world that might otherwise be too dangerous, expensive, or impossible in the real world [5,6]. For example, students can perform laboratory experiments under varying conditions, which would be impossible due to time and costs of physical experiments, and they can explore the intracellular processes that take place in biochemical experiments in real time [7,8]. Moreover, in VR,

students can make mistakes in a controlled environment without any harm (e.g., aversive costs, safety effects) [9,10]. Furthermore, students can progress at their own pace, allowing them to spend more time on particularly relevant issues [11]. Moreover, VR appears to be particularly conducive to affective teaching and learning goals, such as perspective-taking in VR actors [12]. Lastly, students can receive immediate individual feedback from a virtual agent, which is difficult in most classroom settings where one teacher is responsible for assisting several students [13]. These unique characteristics of VR have been associated with several learning affordances as improved spatial knowledge representation, enhanced empathy, increased motivation and student engagement, higher contextualization of learning, and experiential learning scenarios [3,5,14,15]. Thus, VR is particularly relevant for learning content that cannot easily be studied in a traditional classroom setting [16], such as exploring the universe and planetary constellations or visiting the hiding place of Anne Frank in Amsterdam, which is investigated in the current study.

Previous research is often based on the investigation of cause-and-effect relationships that address the direct effects of different technologies on learning outcomes. A systematic review analyzing studies using augmented reality (AR) from high-quality journals revealed that 80% of the studies compare AR to another medium or technology. Only few studies examine how and when learning with AR is effective [17]. However, little research has been conducted not only on unidirectional relationships but on more complex constellations of variables relevant to learning [17,18]. Therefore, the current study aims at elaborating a research design that can be used to investigate both the direct effects of VR technologies and instructional methods as well as the mediating effects of learning process variables latent in VR and the moderating effects of contextual variables.

The paper is structured as follows. First, an overview of the theoretical background is given. Afterward, the methods and results of the study are described. The paper ends with a discussion and a conclusion.

## 2. Theoretical Background

### 2.1. Classification of VR Technology

In everyday language, the term VR is often used as a collective term for heterogeneous technologies and is not used selectively. What they have in common is the understanding of a realistic and computer-generated real-time representation, into which individuals virtually enter, which they experience multisensory and in which they interact via natural and artificial user interfaces. Viewpoint-dependent image generation or egocentrism is also necessary. If motion is caused by the user, then the 3D environment is automatically and immediately displayed from the new perspective [18].

Technologies, collectively known as VR, are advancing. Global players such as *Meta* are driving developments at a rapid pace. It is possible that the following classification may already be outdated in a few years. Nonetheless, the classification of VR based on its technical properties is widespread and appropriate for the current study.

On the one hand, there are 3D environments that are presented via 2D screens [19]. Such VR environments are often classified as non-immersive [20,21]. Stereoscopic representations convey visual depth information to create a spatial impression. However, many stimuli from the real environment, such as other people in the room, or the keyboard, are still perceived. Auditory VR content is presented via speakers or headphones. Interaction and navigation with virtual artifacts are performed via mouse, joystick, or keyboard [22]. When desktop VR (DVR) technology is referred to in the following part of this study, the former description is meant. On the other hand, immersive VR environments are often linked with the use of head-mounted displays (HMDs). Therefore, this technology will be connected with the abbreviation HMD in the following. HMDs enclose ears and eyes. They fill the entire field of vision and largely block out real environmental stimuli. Auditory VR content is conveyed via integrated headphones. The displays present an image to each eye from a slightly different viewing angle. This mimics natural human visual perception, thus creating the stereoscopic impression of a computer-generated virtual world.

The sensors of the HMDs consider head movements and enable perspective changes and movements in space. This is often combined with body suits, gloves, and/or controllers, whose functionalities are used for navigation, selection, and interaction [23,24].

### 2.2. VR as an Educational Technology

VR technologies are considered to have great potential for designing teaching and learning scenarios. They open a range of versatile applications for schools, universities, and other educational institutions [25–27]. Since the 1990s, there has been an increasing effort to use the multiple possibilities of VR to enhance and diversify learning processes in educational contexts [28]. Common fields of VR learning applications are in industry [29,30], in emergency services [31], in medicine [32,33], in teacher education [34,35], and in school contexts [36,37]. Even though an increasing number of studies dealing with VR in educational contexts have been published in recent years [38,39], learning processes are often not mentioned, nor do models or theories form the basis for VR learning scenarios [4,25,40,41]. Rather, the research is technology-driven and often focuses on case studies, anecdotes, and demonstrations of technical prototypes. It should be noted that for VR, there are hardly any specific models and theories for the instructional design of VR learning applications. This could also be shown in two recent studies. Both, an overview study by Radianti and colleagues [18] as well as a meta-analysis by Wu and colleagues [42], pointed out the lack of instructional models.

However, the potential of VR for teaching and learning can be considered on different levels. Dengel and Mägdefrau [43], for example, suggest assessing VR learning outcomes at the levels cognitive, psychomotor, and affective. For acquisition of declarative knowledge in VR (cognitive level), a study compared VR-based instruction with lecture-based instruction and found advantages for the VR group in terms of declarative knowledge acquisition [44]. Similarly, a meta-analysis, which included studies of VR learning applications in nursing, uncovered a moderately large effect in terms of knowledge acquisition in favor of VR [45]. However, a study, in which laboratory safety training is conducted in three different experimental conditions (VR simulation, desktop simulation, paper safety manual), found no meaningful differences in terms of the acquisition of declarative knowledge between the three equivalent content conditions [13]. The empirical findings regarding declarative knowledge acquisition in VR appear to be inconclusive overall. In addition to goals for cognitive learning, the acquisition of procedural knowledge can also be addressed in VR. A review study revealed that VR has been mostly used to teach procedural knowledge [18]. One reason for this is that VR provides optimal conditions for practicing routines by slowing down sequences of actions and practicing as often as desired. Additionally, supporting information for action control and feedback on action execution and results can be integrated. VR learning applications for the acquisition of procedural skills have been used especially where training is rare or not possible. For example, painting of vehicles [46] or behavior in the event of a fire [31] is too infrequently trained. Regarding affective teaching and learning goals, VR experiences open the possibility for learners to empathize with the otherwise distant and seemingly abstract reality of other people's lives, to take on their perspective, and thus to empathize with the situation of these people [12,47]. In a study by Schutte and Stilinovic [15], subjects were shown the realities of a young girl living in a refugee camp either in VR or via a documentary film. The results suggested that, compared to the control condition, the VR experience led to higher levels of empathy for the refugee girl and to a higher willingness to engage in prosocial behavior. In another study, medical staff members were trained in empathic interactions with the elderly. The results showed, but without the use of a comparison group, that VR can improve understanding of age-related health problems and increase empathy for older adults [48]. Several studies also used VR to counteract racism by using VR to increase empathy for individuals who have been discriminated [49,50]. When VR suggests the learner is the person being discriminated against, particularly large effects on empathy are observed [51]. A meta-analysis, which however only included seven articles, examined the relationship between VR and empathy.

The results provided statistically significant positive changes in perspective-taking after VR exposure, but none in empathy [52]. A review by Martingano et al. [47], which included 43 studies, demonstrated that VR improved emotional empathy but not cognitive empathy. Consequently, VR can evoke compassionate feelings but does not appear to encourage imagining other people's perspectives. Further analysis of the study found that VR was no more effective at promoting empathy than less technologically advanced interventions (e.g., reading an article).

From a learning psychology perspective, it is not enough to assess learning activities based on learning outcomes. Rather, the latent processes that happen during a VR experience are of interest. These are central for a deeper understanding of learning in VR. Thus, the scientific investigation of such processes makes it possible to describe the quality of learning experiences in VR in more detail. Therefore, if the potential of VR for educational scenarios is to be assessed, the underlying learning processes should also be considered. Experiencing presence [53] and flow [54] have been shown to be promising mediating factors. These constructs, which are frequently discussed in the literature in the context of VR and learning, will therefore be described in detail below.

### 2.3. Mediating Variables: Presence and Flow

A central phenomenon that is repeatedly associated with VR is the feeling of being present in a VR environment [53]. In this context, the illusion of being present, i.e., being immersed and merged with the virtual world [55], is often understood as a psychological perceptual process that can vary between individuals [56]. A distinction is often made between physical presence, the feeling of being localized at another place, social presence, the feeling of being there with other virtual or real actors, and self-presence, the feeling of actually being there as an individual [57–59]. The relationship between experiencing presence in VR and learning has already been investigated in several studies. Many researchers assume that the use of technologies that are considered to be highly immersive (e.g., HMDs) leads to more presence experience than the use of technologies described as less immersive (e.g., DVR) [60]. In a study, Stevens and Kincaid compared HMD-based VR with a desktop application, both dealing with military security training. They demonstrated that significantly more presence was experienced using HMDs and that performance in a subsequent shooting exercise was also better than using desktops [61]. Additionally, Krokos and colleagues showed in a study involving spatial recall that subjects recalled significantly more using an HMD than subjects who used a comparable desktop application [62]. Further study results also indicate that the experience of presence has a positive effect on the learning process to the extent that a higher level of presence experience requires a stronger focus of attention on learning-relevant stimuli [43,63]. In the literature, however, opposite findings are detectable. Makransky and colleagues, for example, found a negative correlation between learning and presence experience. Biochemical processes were perceived either via desktop-based or HMD-based VR. When using HMDs significantly more presence was experienced. However, the learning results were partly worse than using desktops. The authors concluded that higher presence could lead to distraction by many irrelevant details or high arousal [64]. In summary, the evidence is inconclusive, although a large proportion of the study results suggests that more presence can support learning in VR. Therefore, the present study is dedicated to exploring the mediating effect of presence experience on learning.

The experience of presence in VR correlates with the experience of flow in VR [65]. Flow experience is assumed to mean that one experiences an activity as gratifying. Thereby, no or hardly any separation between him- or herself and the activity is perceived [66]. In addition, the course of the action is automated, and the action is usually performed faster and more effectively. A further characteristic of the flow is the loss of the sense of time [54]. Rheinberg and colleagues consider flow as a multidimensional construct consisting of the facets absorbedness and smooth automated progression. This refers to the complete absorption in an activity and the flowing of chains of action [67]. The experience

of flow is also correlated with the success of a VR learning activity. Bodzin and colleagues investigated students who explored the watershed of their city's river. The students reported that they had experienced more flow and learned more. However, there was no comparison group implemented [68]. In a further study, engineering students were trained in VR in order to care for vehicle parts adequately. Procedural skills tested in a post hoc test could be significantly explained by experiencing flow in VR. Again, no control group was implemented [69]. Another study analyzed a performance-based rhythm game, either played using an HMD or a laptop. Using HMDs resulted in a stronger sense of presence than playing on laptops. Additionally, the experience of flow could significantly explain variance of game results. However, the HMD and laptop group did not differ with respect to flow [54]. In general, there is still little empirical research on the association of flow experience in VR and learning parameters. The present study ties in with this.

*2.4. Moderating Variables: Prior (Technical) Knowledge*

Learning outcomes are always dependent on learner characteristics. In VR, it seems that prior knowledge about the topic as well as prior technical knowledge in particular are determinants of learning success that have to be controlled. If these are left unconsidered, learning outcomes may be negatively impacted [70,71]. Here, prior knowledge refers to learners' knowledge base and prior technical knowledge to the previous experiences with the technology. VR learning applications generally carry the risk of getting lost in the variety of possibilities. The learning process in VR itself already ties up processing capacities. In the absence of prior (technical) knowledge, additional capacities are required to process the learning material appropriately. Especially with learners who are not very familiar with VR technologies or who do not know anything about the topic, cognitive overload is likely to occur, and it is recommended to use instructional strategies in order to create capacities for the relevant learning activities. For example, Meyer et al. tested the principle of pre-training in VR by presenting half of the participants an image of the structure of a biological cell before they explored the cell and its components in VR. Learning-enhancing effects on memory and transfer performance were shown [70]. It is noteworthy that the use of certain instructional aids for people with little prior knowledge (novices) may facilitate learning, while the same methods do not support or even hinder the learning of persons with high prior knowledge (experts). This interaction is called the expertise reversal effect. Kalyuga and Renkl [72] explain this effect based on the Cognitive Load Theory [73]. The working memory of experts is additionally burdened rather than relieved by the use of instructional support. Processing redundant information can disrupt learning processes. A high degree of expertise thus limits the range of application of instructional methods. Therefore, in this study, both the prior knowledge and the technical prior knowledge are assessed as contextual control variables in order to determine interactions with the instructional methods used. With regard to the structuring of the learning process, experienced learners with prior (technical) knowledge are more likely to benefit from learning scenarios that encourage exploration, whereas exposition-based learning scenarios are more suitable for inexperienced learners without prior (technical) knowledge.

**3. Research Aim and Hypotheses**

Many studies in the field of educational technology are driven by the need to prove or disprove the effectiveness of certain technologies. Nevertheless, so-called media comparison studies are often criticized [74,75]. Parong and Mayer [76] refer to the comparison of two media methods as an "apples-to-oranges type of comparison" (p. 789), as underlying instructional methods are neglected. Therefore, in this study, a research design is used that considers two VR technologies, two instructional methods, two learning processes, as well as two contextual variables.

As a basis for this study, the Cognitive Affective Model of Immersive Learning (CAMIL) by Makransky and Petersen [38], the Educational Framework for immersive Learning (EFiL) by Dengel and Mägdefrau [43], and the Meaningful immersive VR Learn-

ing (M-iVR-L) model by Mulders, Buchner, and Kerres [77] were used. These three recent models, especially the CAMIL, are the basis for the research design and the experimental study presented in this paper. Figure 1illustrates the research design on which the experimental study is built.

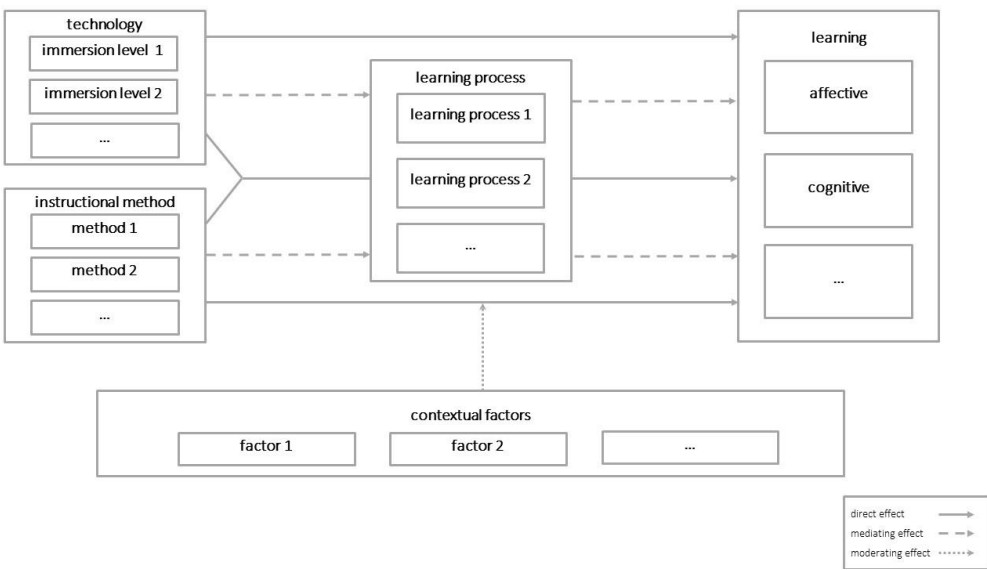

**Figure 1.** Research design.

In the research design, technologies with different levels of immersion and instructional methods are integrated as independent variables. The main effects of the independent variables will be investigated regarding the dependent variables, i.e., the indicators of learning at various levels. Additional variance can be identified if not only the main effects of the independent variables on the dependent variables are considered, but also the interactions between the independent variables are analyzed (technology × method) [38].

By integrating mediator and moderator variables into the research design, in addition to the consideration of independent and dependent variables, the simultaneous consideration of several third-party variables is possible. By controlling for such third-party variables, confounding and spurious effects can be avoided. Studies that were exclusively dedicated to the comparison of two forms of media presentation (e.g., HMD-VR vs. DVR) often failed to control for person-related variables. If additional variables are integrated that, based on previous research results, seem to be related to the independent and dependent variables, on the one hand biased results can be avoided, and at the same time additional variance of the dependent variables can be systematically explained.

In the current study, the research design is transformed into a 2 × 2 experiment. The aim of the study is to investigate the extent to which HMD-VR and DVR are suitable to significantly support the learning of middle school students. Simultaneously, two instructional methods are tested. Exposition and exploration are contrasted as instructional methods [78]. The implementation of the exposition method is guided by the principle of direct instruction [79,80] and the design principle of segmentation [81]. Students in the exposition conditions were given specific instructions that guided them through the VR experience. They followed a determined learning path. Accordingly, learning was sequential and in segments appropriate for learning. In small steps, students were guided through the VR environment (e.g., *"First, look around. Then turn to your left to face the desk."*). Feedback on learning progress was given (e.g., *"Now you have already explored half of the rooms."*). Students who experienced the VR scenario exploratively were instructed to discover the VR environment freely and self-directedly. No further instructions were given.

Using the example of the historically relevant VR environment *Anne Frank VR House*, namely the in VR recreated hiding place of Anne Frank, a girl of Jewish heritage during

the Second World War, in Amsterdam, cognitive (i.e., declarative knowledge) and affective (i.e., perspective-taking in Anne) parameters of learning and the evaluation of the VR application (i.e., satisfaction, recommendation) are analyzed. A particular focus of the study will be on learning-relevant processing (here: the experience of presence and flow) and contextual variables (here: prior (technical) knowledge) that can adequately explain the learning results.

In this study, the following hypotheses are tested, with the mediating and moderating effects being considered as particularly interesting. The phrase learning outcomes always refers to affective and cognitive learning indicators as well as the evaluation of the VR environment. Hypotheses one and two test main effects, hypothesis three tests moderation effects, and hypothesis four tests mediation effects.

1.  Learning outcomes differ depending on the VR technology.
    a.  regarding affective learning indicators: HMD-VR > DVR.
    b.  regarding cognitive learning indicators: HMD-VR = DVR.
    c.  regarding evaluation: HMD-VR > DVR.
2.  Learning outcomes differ depending on the instructional method (exploration ≠ exposure).
3.  Prior (technical) knowledge affects the learning outcomes depending on the instructional method.
    a.  The more prior (technical) knowledge is available, the more conducive to learning outcomes in the exploration experimental condition (and reverse).
    b.  The less the prior (technical) knowledge, the more conducive to learning outcomes in the exposition experimental condition (and reverse).
4.  Learning process variables convey the relationships between VR technology and learning outcomes.
    a.  More presence is experienced in the HMD-VR conditions than in the DVR conditions. The more presence is experienced, the more conducive to learning outcomes.
    b.  More flow is experienced in the HMD-VR conditions than in the DVR conditions. The more flow is experienced, the more conducive to learning outcomes.

## 4. Methods

### 4.1. Participants

The sample consisted of 132 middle school students in Germany (65 female, 63 male, 4 non-binary). The sample is composed of students aged 12 to 17 years ($M = 13.84$, $SD = 0.92$). A total of 55% reported attending the eighth grade and another 45% ninth grade. Students were recruited using flyers and social media posts. They were randomly divided into four experimental conditions: HMD-VR and exposition ($N = 37$), HMD-VR and exploration ($N = 37$), DVR and exposition ($N = 31$), and DVR and exploration ($N = 27$).

### 4.2. VR Environment

The VR application *Anne Frank VR House* offers the exploration of the hiding place of Anne Frank at the time of the Second World War in Amsterdam. It is a freely available application, jointly developed by Force Field VR and the Anne Frank Foundation. The everyday reality of those living in the hiding place is presented in detail, without showing any of the decedents personally. Between 1942 and 1944, the hiding place was the home of 13-year-old Anne, her sister Margot, her parents, and four other Jewish people. Additional information about the life in hiding, the history of those in hiding, and holocaust in general is included. An insight is offered by Figures 2 and 3.

The *Anne Frank VR House* can be explored by two VR technologies. On the one hand, HMDs and two linked controllers can be used. Teleportation and head movements make it possible to discover the hiding place room by room. On the other hand, the virtual hiding place can be explored by various devices with 2D screens (e.g., tablets, laptops) by accessing

the 360° web application with equivalent content. Here, the field of view is changed either by the mouse cursor or by the touch function.

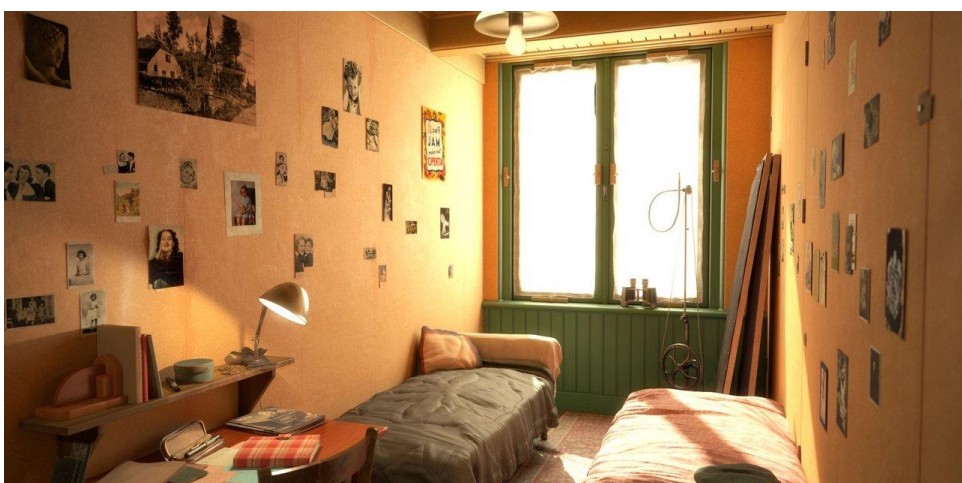

**Figure 2.** Room of Anne Frank and Fritz Pfeffer. https://www.annefrank.org/en/about-us/news-and-press/news/2018/6/12/anne-frank-house-vr-launched/ (accessed on 12 February 2023).

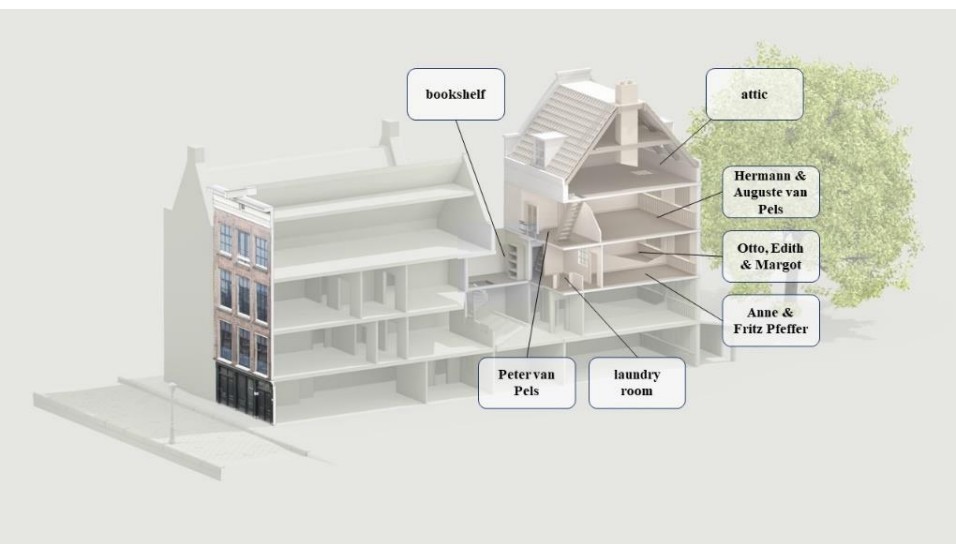

**Figure 3.** The hiding place.

### 4.3. Procedure

The study was conducted between May and December 2021. The exact procedure of the study is described below. Furthermore, an illustration of the procedure can be found in Figure 4. Students were first given an online pre-questionnaire including questions about their prior (technical) knowledge and then were randomly assigned to one of the four conditions. All participants then either completed the HMD-VR or DVR simulation with additional guidance or self-directed based on verbal or written step-by-step instructions. An *Oculus Quest 2* was used in the HMD-VR conditions, and a *Dell Latitude 3510* in the DVR conditions. The VR experience itself took 30 min on average. Lastly, participants were given an online post-questionnaire that included items regarding sociodemographic variables, knowledge acquisition, perspective-taking, evaluation, as well as questionnaires for the learning process variables.

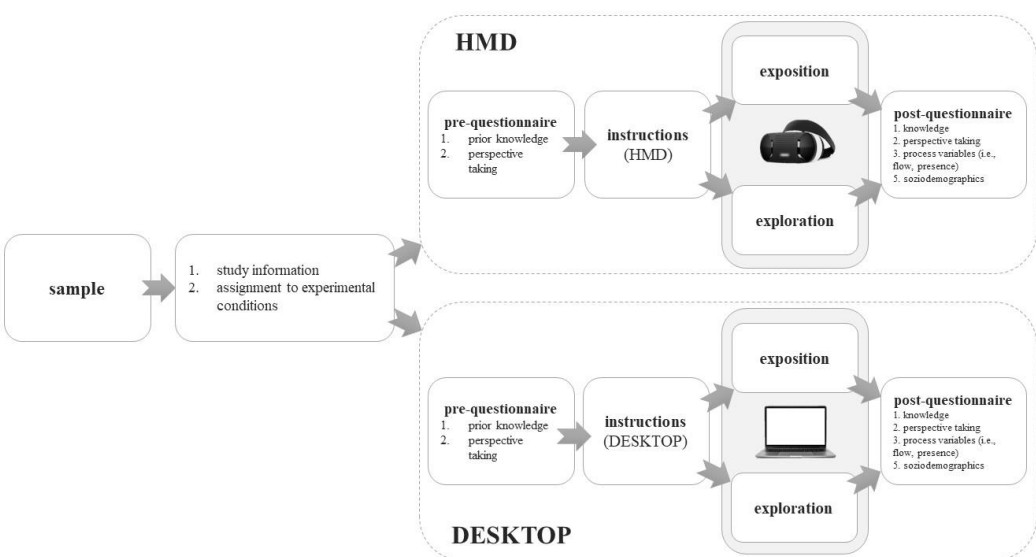

**Figure 4.** Study process (illustration based on [82]).

*4.4. Measurements*

Data were collected at two measurement points. Gender, age, and grade level were assessed as sociodemographic variables. Learning outcomes were collected at various levels. To measure knowledge acquisition (i.e., cognitive learning), the students should answer ten knowledge questions (e.g., "*What did Anne's diary look like?*"). A maximum of two points per question was reachable, resulting in a maximum point value of 20. The internal consistency of the knowledge test is $\alpha = 0.71$. Moreover, students assessed their knowledge level on a scale from 1 ("*not available*") to 10 ("*extensive knowledge*"). Students should answer this question before and after exploring the VR. The first measurement also served to record prior knowledge regarding the topic Anne Frank. Next to cognitive learning objectives, affective learning was examined, too. As an affective learning goal, perspective-taking in Anne Frank was determined. Perspective-taking is understood as the capacity to empathize with another person's feelings. Perspective-taking is a prerequisite for the emergence of empathy or compassion [83,84]. To measure the extent of perspective-taking, students should assess how well they could empathize with Anne Frank's living conditions in the hiding place. A ten-point scale from 1 ("*not at all*") to 10 ("*very well*") was used. This question was asked twice, before and after the VR experience. Additionally, the students were asked to evaluate their VR experiences. To do so, they should assess their level of satisfaction ("*I am satisfied with the virtual learning experience.*") as well as their tendency to recommend it ("*I would recommend the application to other students.*") on a scale of 1 ("*strongly disagree*") to 5 ("*strongly agree*").

In addition to recording prior knowledge about Anne Frank, prior experiences with the technology were surveyed as a control variable with one closed question ("*Have you already had experience with VR?*").

The Multimodal Presence Scale (MPS) was used to examine the experience of presence [59,85]. The self-report scale originally included 15 items. The dimensions physical presence, self-presence, and social presence consist of five items each. In this study, the dimension of social presence was omitted since it is no virtual environment in which the student could meet other virtual or real actors. The underlying five-point scale ranges from 1 ("*do not agree at all*") to 5 ("*completely true*"). Cronbach's alpha in the studies by Volkmann et al. [85] for physical presence ranges between $\alpha = 0.69$ and $\alpha = 0.82$ and for self-presence between $\alpha = 0.84$ and $\alpha = 0.89$. The internal consistencies in this study are $\alpha_{physical} = 0.88$ and $\alpha_{self} = 0.93$. Volkmann et al. [85] further report norm values from studies in which they manipulated the level of interactivity with a virtual agent. Mean values ranged from $M = 2.93$ ($SD = 0.96$) to $M = 3.01$ ($SD = 0.85$) for self-presence and between

$M = 3.15$ ($SD = 0.88$) and $M = 3.54$ ($SD = 0.71$) for physical presence. An exemplary item for physical presence is "*The virtual environment seemed real to me.*" and for self-presence "*It felt like my real hand was in the virtual environment.*".

To measure the experience of flow, the Flow Short Scale (FKS) was used [67]. The instrument consists of the two facets of smooth automated flow (example: "*I was completely absorbed in what I was doing.*") and absorbedness (example: "*I had no trouble concentrating.*") and a total of ten items. The questionnaire has a seven-point scale as a response format from 1 ("*strongly disagree*") to 4 ("*partly agree*") to 7 ("*strongly agree*"). Rheinberg and colleagues named values from comparative studies. A mean of $M = 5.16$ ($SD = 0.93$) was found for graffiti spraying and a mean of $M = 4.57$ ($SD = 1.13$) for solving a statistics task. The authors reported a Cronbach's alpha of $\alpha = 0.90$ [67]. The internal consistency in the current study was $\alpha = 0.81$.

### 4.5. Statistical Analyses

In addition to simple descriptive statistical analyses, a multivariate analysis of variance (MANOVA) and post hoc ANOVAs were conducted with technology and instructional method as the independent variables and the various learning indicators as the dependent variables. To explore the mediating effects of the learning process variables, flow, and presence, mediation analyses were calculated. In addition, moderation analyses were conducted to reveal possible interactions with prior (technical) knowledge. For these analyses, SPSS and the macro PROCESS [86] were used.

## 5. Results

The presentation of the statistical analyses is divided into four main sections. First, the descriptive statistics of the variables collected are reported. In the next step, the testing of hypotheses one and two is adequately presented in the context of a MANOVA. The moderation analyses concerning hypothesis three can be found in the following section. Finally, to answer hypothesis four, $t$-tests for independent samples, correlation, and mediation analyses are presented.

### 5.1. Descriptive Characteristics of the Sample

Table 1 shows the measurement points, ranges, sample sizes, means, standard deviations, and Cronbach's alphas as a measure of internal consistency for each scale or item.

**Table 1.** Descriptive characteristic values of all scales used in the study. Annotations: MP = measurement point, $N$ = sample size, $M$ = mean, $SD$ = standard deviation, $\alpha$ = Cronbach's alpha.

| Variable | MP | Range | $N$ | $M$ | $SD$ | $\alpha$ |
|---|---|---|---|---|---|---|
| knowledge | pre | 1–10 | 132 | 4.16 | 2.17 | - |
| | post | 1–10 | 132 | 6.79 | 1.68 | - |
| knowledge test | post | 0–20 | 132 | 10.89 | 4.36 | 0.71 |
| perspective-taking | pre | 1–10 | 132 | 4.77 | 2.40 | - |
| | post | 1–10 | 132 | 6.67 | 2.13 | - |
| flow | | | | | | |
| smooth automated flow | post | 1–7 | 128 | 5.18 | 1.36 | 0.68 |
| absorbedness | post | 2–7 | 128 | 5.37 | 1.38 | 0.85 |
| presence | | | | | | |
| physical presence | post | 1–5 | 124 | 3.35 | 0.99 | 0.88 |
| self-presence | post | 1–5 | 124 | 2.82 | 1.14 | 0.93 |
| evaluation | | | | | | |
| satisfaction | post | 1–5 | 124 | 4.31 | 0.91 | - |
| recommendation | post | 1–5 | 124 | 4.33 | 0.93 | - |

Concerning prior technical knowledge, in the HMD-VR experimental conditions, 70% reported to previously had experience with VR, whereas in the DVR experimental conditions, 59% reported previous experience with VR.

Regarding the instruments used to operationalize the learning outcomes from the pre- to the post-measurement point, the following can be stated: Across all experimental conditions, from the first to the second measurement point, there is an increase in knowledge and perspective-taking into Anne Frank's situation (see Figure 5).

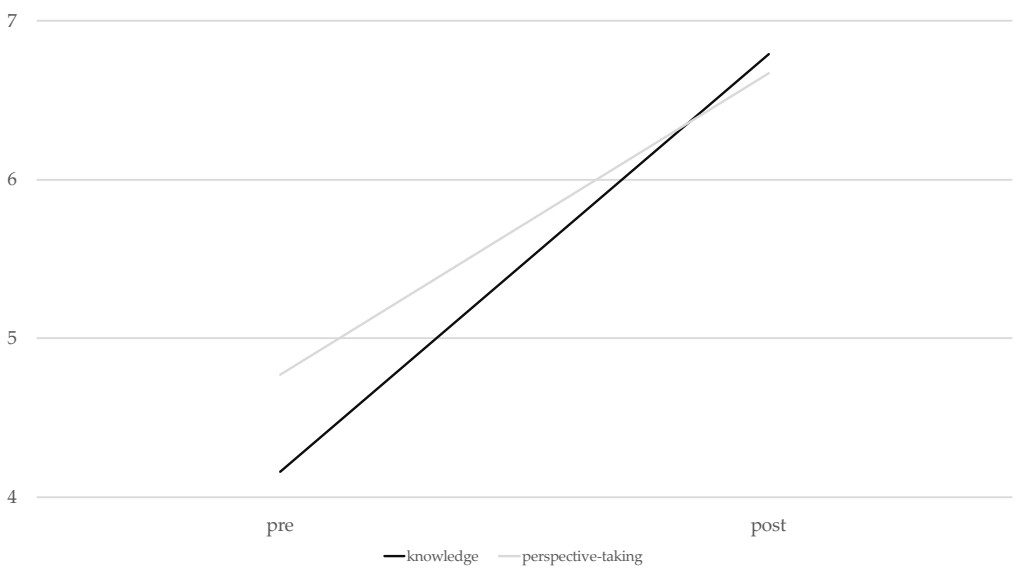

**Figure 5.** Knowledge and perspective-taking: pre to post.

In the knowledge test, the students received slightly more than half of the maximum twenty points to be achieved on average. However, a very high dispersion can be observed. The performance of the students in the knowledge test is therefore very different. Considering the overall evaluation of the VR application, high mean values and low dispersion are detected averaged over all experimental conditions. Just a few outliers were not satisfied and would not recommend the VR application to others. Regarding the measuring instruments used to operationalize learning processes (i.e., presence, flow), the following can be reported: The experience of flow averaged over all four experimental conditions and compared to the values reported by Rheinberg et al. for particularly absorbing and smooth activities such as graffiti spraying [67] can be classified as very high. Particularly the absorbedness reached very high values on average. Accordingly, the students seem to have been captivated by the VR experience across all media and methods conditions. Moreover, the experience of presence is to be classified as high, averaged over the experimental conditions. This applies to both subscales. Consequently, the students felt present in Anne's hiding place and experienced themselves as acting persons within the VR environment. The results are comparable to those reported by Volkmann et al. [85].

*5.2. Main Effects*

To test hypotheses one and two, a two-way MANOVA was used. As independent variables, both the technology and the instructional method were integrated. As dependent variables, the knowledge test and both items for the overall evaluation were used. In addition, two new variables are calculated to operationalize learning. Both perspective-taking and knowledge were queried at both measurement points by two single items. To measure the increase in perspective-taking and knowledge from the first to the second measurement time, two new variables were calculated that allow a pre-post comparison. As already shown in Table 1 and Figure 5, the values at the second measurement are numerically higher than at the first measurement. Therefore, the pre-value was subtracted from the

post-value. Positive values on these two new items mean an increase in perspective-taking and knowledge, negative values a decrease. On average, knowledge improved by 2.63 units across all test conditions ($SD = 2.25$) and the perspective-taking by 1.90 units ($SD = 2.37$). The two new items were also included as dependent variables into the MANOVA.

The MANOVA showed statistically significant differences between the stages of variable technology ($F(5, 108) = 8.67$, $p < 0.001$, partial $\eta^2 = 0.39$, Wilk's $\Lambda = 0.61$) and between the stages of the variable instructional method ($F(5, 108) = 2.68$, $p < 0.001$, partial $\eta^2 = 0.17$, Wilk's $\Lambda = 0.83$) for the combined dependent variables. Thus, technology is the factor in the MANOVA that can explain the most variance (39%), following instructional method that can explain 17%. The interaction between technology and instructional method has not become significant ($F(5, 108) = 0.82$, $p = 0.59$, partial $\eta^2 = 0.06$, Wilk's $\Lambda = 0.94$). Since significant findings were found for both factors of the analysis, post hoc multifactorial ANOVAs for each dependent variable were calculated. The Bonferroni correction was used to decrease the risk of a type I error when making multiple statistical tests. Table 2 shows the results of the independent variable technology, Table 3 the results of the independent variable instructional method.

**Table 2.** Results of ANOVAs for the independent variable technology. Annotations: $F$ = test statistic, $df$ = degrees of freedom, $p$ = probability, $\eta^2$ = effect size partial eta$^2$, significance level * $p < 0.05$, ** $p < 0.01$, *** $p < 0.001$.

| Dependent Variable | F | df | p | $\eta^2$ |
|---|---|---|---|---|
| knowledge (pre-post comparison) | 0.07 | 1, 115 | 0.80 | 0.00 |
| knowledge test | 21.90 | 1, 115 | 0.00 *** | 0.16 |
| perspective-taking (pre-post comparison) | 3.23 | 1, 115 | 0.08 | 0.03 |
| satisfaction | 4.85 | 1, 115 | 0.03 * | 0.04 |
| recommendation | post | 1, 115 | 0.01 ** | 0.06 |

**Table 3.** Results of ANOVAs for the independent variable instructional method. Annotations: $F$ = test statistic, $df$ = degrees of freedom, $p$ = probability, $\eta^2$ = effect size partial eta$^2$, significance level * $p < 0.05$, ** $p < 0.01$.

| Dependent Variable | F | df | p | $\eta^2$ |
|---|---|---|---|---|
| knowledge (pre-post comparison) | 6.94 | 1, 115 | 0.01 ** | 0.06 |
| knowledge test | 3.82 | 1, 115 | 0.05 * | 0.03 |
| perspective-taking (pre-post comparison) | 0.81 | 1, 115 | 0.37 | 0.01 |
| satisfaction | 0.14 | 1, 115 | 0.71 | 0.00 |
| recommendation | 1.69 | 1, 115 | 0.20 | 0.01 |

Hence, a significant main effect for the variable technology was found averaged across all learning indicators. However, a superiority of HMD-based VR was uncovered only for two evaluative indicators, and for one affective learning indicator, even if not statistically significant. The results for the knowledge test are different: Here, the students in the DVR conditions performed significantly better. Consequently, subhypothesis a could rather not be supported by the data of this study. With respect to the acquisition of knowledge (subhypothesis b), the findings are ambiguous. As expected, no differences between the two forms of technology (DVR vs. HMD-VR) could be found for one of the cognitive learning indicators. The assumed null hypothesis would be accepted. The situation was different for the indicator knowledge test. Here, there was an advantage for those who were in the DVR conditions. This difference was highly significant. The difference between the two groups is approximately one standard deviation unit. This means that, on average, the students in the DVR conditions solved two more questions correctly than students in the HMD-VR

conditions. The significant difference between the two groups could possibly be explained by the fact that cognitive capacities that were tied up by many environmental details and by the high interactivity of the HMD-VR technology were free in the DVR conditions, and the focus could be placed on the acquisition of expertise [64,87]. As expected according to subhypothesis c, students in the HMD-VR experimental conditions were significantly more satisfied (4% variance explanation) and would be more likely to recommend the application to classmates (6% variance explanation). It can be assumed that this finding is also partly due to the novelty effect [88,89]. This effect was certainly more pronounced among students in the HMD-VR conditions than in the DVR conditions. Nevertheless, learning with HMDs seems to have been more emotionally engaging and motivating for the students. According to recent assumptions of motivation theories [90,91], multimedia learning environments trigger, when they are perceived as appealing, a high initial situational interest, which in turn can have a positive effect on subsequent learning processes.

The main effect for the variable instructional method was also significant averaged across all learning indicators. With respect to two cognitive learning indicators, performance was significantly better in the exposition conditions than in the exploration conditions. Hence, students in the exposition conditions solved more questions correctly in the knowledge test and estimated their knowledge gain from the first to the second measurement point as greater. In all, 6% and 3% of the variance in the two learning indicators can be explained by the instructional method, respectively. The differences in the knowledge test can possibly be explained by considering that the students in the exposition conditions, through the instructions given, encountered the content that was later asked in the knowledge test. All of the VR content was explored in the exposition conditions. It was unlikely to have missed relevant content. Students in the exploration conditions, on the other hand, were free to choose what content they wanted to engage with and to what intensity. Often in the exploration conditions, certain elements were examined in a time-consuming manner, while others were neglected, often also depending on personal interests. Consequently, the undirected second hypothesis can only be supported to a limited extent, insofar as the learning outcomes of the two experimental conditions differ only for two indicators. In both significant findings, however, the same tendency can be seen that students in the exposition group have an advantage over students in the exploration group with respect to knowledge acquisition.

### 5.3. Moderating Effects

In the current study, it has been assumed that prior knowledge about the topic and prior technical knowledge determine the relationship between the independent variable instructional method and the learning outcomes as dependent variables. In order to adequately check the assumed moderating effects of the two control factors, the PROCESS macro for SPSS was used [86]. Linear regressions were performed. In the regressions, as independent variable the instructional method and as moderators prior knowledge about Anne Frank and prior technical knowledge were included. For each learning indicator (i.e., knowledge test, knowledge pre-post comparison, perspective-taking pre-post comparison, satisfaction, recommendation), one regression has been calculated. Table 4 provides an overview of the results. For each dependent variable, the statistics on the interactions of instructional methods and the two control variables are shown.

The majority of statistical analyses could not find any moderating effects of the control factors. Only for one learning indicator (i.e., the subjective assessment of knowledge gain), the interaction between the prior knowledge and the instructional method could significantly precede the learning outcomes. In all, 1.6% of the total variance could be explained by the interaction. However, further analysis with scatterplots showed that while the effects were in the expected direction (expertise reversal effect [72]), they were not significant. Accordingly, individuals with much prior knowledge are more likely to benefit in the exploration conditions, while those with little prior knowledge are more likely to benefit in the exposition conditions. However, the majority of the results indicate that prior

knowledge is insignificant in determining whether someone is other- or self-directed in exploring a VR environment. Hypothesis three cannot be supported.

**Table 4.** Results of the moderation analyses. Annotations: $F$ = test statistic, $df$ = degrees of freedom, $p$ = probability, $\Delta R^2$ = effect size delta $R^2$, significance level * $p < 0.05$.

| | $F$ | $df$ | $p$ | $\Delta R^2$ |
|---|---|---|---|---|
| **knowledge (pre-post comparison)** | | | | |
| prior knowledge | 4.38 | 1, 126 | 0.04 * | 0.0163 |
| prior technical knowledge | 1.66 | 1, 126 | 0.20 | 0.0052 |
| **knowledge test** | | | | |
| prior knowledge | 0.02 | 1, 126 | 0.88 | 0.0002 |
| prior technical knowledge | 2.38 | 1, 126 | 0.13 | 0.0178 |
| **perspective-taking (pre-post comparison)** | | | | |
| prior knowledge | 0.43 | 1, 126 | 0.51 | 0.0034 |
| prior technical knowledge | 0.03 | 1, 126 | 0.86 | 0.0003 |
| **satisfaction** | | | | |
| prior knowledge | 0.06 | 1, 118 | 0.80 | 0.0006 |
| prior technical knowledge | 0.03 | 1, 118 | 0.87 | 0.0002 |
| **recommendation** | | | | |
| prior knowledge | 0.00 | 1, 118 | 0.98 | 0.0000 |
| prior technical knowledge | 0.18 | 1, 118 | 0.67 | 0.0015 |

*5.4. Mediating Effects*

In the last section of the results report, the focus is on hypothesis four dealing with mediation analyses. A mediator explains the relationship between an independent and a dependent variable. Thus, a mediation analysis is also an analysis of causal effects. In addition to direct effects, indirect effects via third-party variables are examined. According to Baron and Kenny, four assumptions for the existence of a mediation must be fulfilled: First, between the independent variable and the dependent variable, there is a direct relationship (path c). This path is also called the total effect. Second, the independent variable must correlate with the mediator (path a). Third, the mediator and the dependent variable must be connected (path b). Fourth, in full mediation, the direct path between independent and dependent variable loses its significance (path c'). If the fourth step is not fulfilled, it is referred to as partial mediation [92]. Figure 6 represents the principles of mediation analyses.

In this study, it has been assumed that latent processes (i.e., presence, flow) while learning in VR can significantly explain the relationships between technology and learning indicators.

The first assumption of Baron and Kenny, namely the direct relationship between technology and the various learning indicators (path c), has already been sufficiently verified by the MANOVA and the subsequent ANOVAs in Section 5.2. The main effect technology for the combined dependent variables was significant.

To test the second assumption (path a), the relationships between technology and learning process variables were investigated. For this purpose, multiple *t*-tests for independent samples were carried out. Here, the method tests whether there are differences in the feeling of presence and flow regarding the stages of technology (HMD-VR vs. DVR). The results of the *t*-tests are shown in Table 5. A positive mean difference indicates that students in the HMD-VR conditions achieved higher values than students in the DVR conditions.

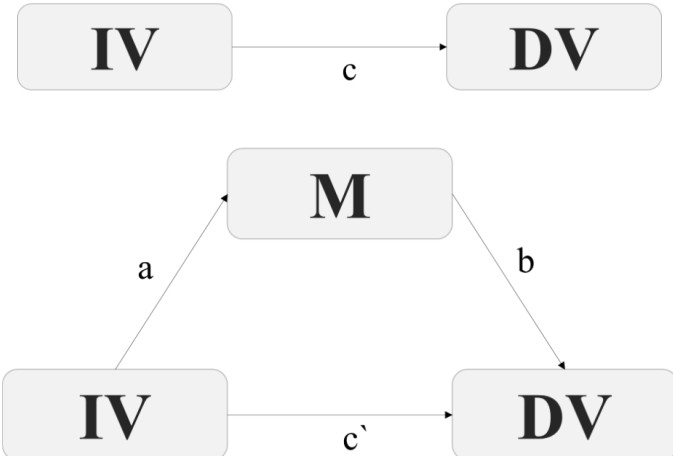

**Figure 6.** Mediation principles (illustration based on [92]). Annotations: IV = independent variable, DV = dependent variable, M = mediator.

**Table 5.** Results of the multiple *t*-tests for independent samples for presence and flow at the stages of technology. Annotations: *MD* = mean difference, *SD* = standard deviation, *df* = degrees of freedom, *t* = test statistic, *p* = probability, *d* = effect size *d*, significance level *** $p < 0.001$.

| Mediator | *MD* | *SD* | *df* | *t* | *p* | *d* |
|---|---|---|---|---|---|---|
| flow | | | | | | |
| smooth automated flow | 1.17 | 0.23 | 103 | 5.17 | 0.00 *** | 0.95 |
| absorbedness | 1.75 | 0.19 | 126 | 9.18 | 0.00 *** | 1.64 |
| presence | | | | | | |
| physical presence | 1.06 | 0.15 | 108 | 6.88 | 0.00 *** | 1.26 |
| self-presence | 1.06 | 0.18 | 111 | 0.13 | 0.90 | 1.05 |

Regarding the learning processes, statistically highly significant differences could be found between the HMD-VR group and the DVR group. Students in the HMD-VR conditions experienced more flow. This applies to both subscales. Regarding the physical presence subscale, the students of the HMD-VR group felt more presence than those of the DVR group, but not regarding self-presence.

To test the third assumption of Baron and Kenny (path b), descriptive correlation analyses were used. The correlative relationships were determined due to a lack of normal distribution with the nonparametric Spearman rank correlation. The results of the correlation analyses are shown in Table 6. Various statistically significant correlations could be determined. Thus, the experience of flow seems to be related to increased satisfaction, greater tendency to recommend, and better performance in the knowledge test. The presence experience also correlates significantly positively with satisfaction and recommendation and, depending on the subscale, positively with the performance in the knowledge test.

In the following, the fourth assumption of Baron and Kenny and thus the mediation assumptions are checked inferential statistically. In the mediation model, technology was included as an independent variable, the two subscales of flow and physical presence as mediator variables, and satisfaction, recommendation, and the knowledge test as dependent variables. The self-presence subscale was excluded due to insufficient correlations with the technology and the learning indicators. Based on the results of the MANOVA and ANOVAs, the newly formed variables for pre-post comparison were also excluded, as no significant associations with the technology could be found. As a result, a total of nine mediation models were examined. Like the moderation analyses, the mediation analyses were also carried out using Hayes' macro PROCESS and linear regressions [86].

**Table 6.** Spearman rank correlations between learning process variables and learning indicators. *Annotations:* KC = knowledge pre-post comparison, PC = perspective-taking pre-post comparison, KT = knowledge test, Sa = satisfaction, Re = recommendation, significance level * $p < 0.05$, ** $p < 0.01$.

| | Learning Indicator | | | | |
| --- | --- | --- | --- | --- | --- |
| **Mediator** | **KC** | **PC** | **KT** | **Sa** | **Re** |
| flow | | | | | |
| smooth automated flow | 0.11 | 0.16 | 0.22 ** | 0.47 ** | 0.36 ** |
| absorbedness | 0.15 | 0.11 | 0.23 ** | 0.41 ** | 0.36 ** |
| presence | | | | | |
| physical presence | 0.09 | 0.03 | −0.15 | 0.48 ** | 0.48 ** |
| self-presence | 0.01 | 0.04 | 0.22 * | 0.40 ** | 0.40 ** |

For example, a statistically significant mediation could be detected for the mediator flow (here: smooth automated flow) and the dependent variable satisfaction. Within the mediation analysis, a total effect of technology on satisfaction was uncovered (c = −0.58 **), directed in such a way that those in the HMD-VR group were significantly more satisfied than those in the DVR group. After the mediator was included in the model, the technology could predict the mediator, i.e., flow, significantly (a = −1.19 ***). Students in the HMD-VR condition experienced significantly more flow than students in the DVR condition. Flow, in turn, predicted the satisfaction of the students (b = 0.29 ***). The more flow was experienced, the more satisfied the students were. The path c', i.e., the direct effect of technology on satisfaction, was not significant (c' = −0.23). Therefore, a full mediation could be established. After the mediator flow was added to the analysis, the correlation between technology and satisfaction lost its statistical significance. Consequently, there is no statistically significant direct link between technology and satisfaction, but an indirect connection that can be explained by flow. Accordingly, the relationship between technology and satisfaction is fully mediated by experiencing flow (indirect effect ab$_{\text{smooth automated flow}}$ = −0.38; 95% CI [−0.571, −0.169]). An illustration of the causal relationships is shown in Figure 7. Overall, in four out of the nine analyses carried out, a significant full mediation could be revealed (2x physical presence, 2x smooth automated flow), but only for the evaluative learning indicators.

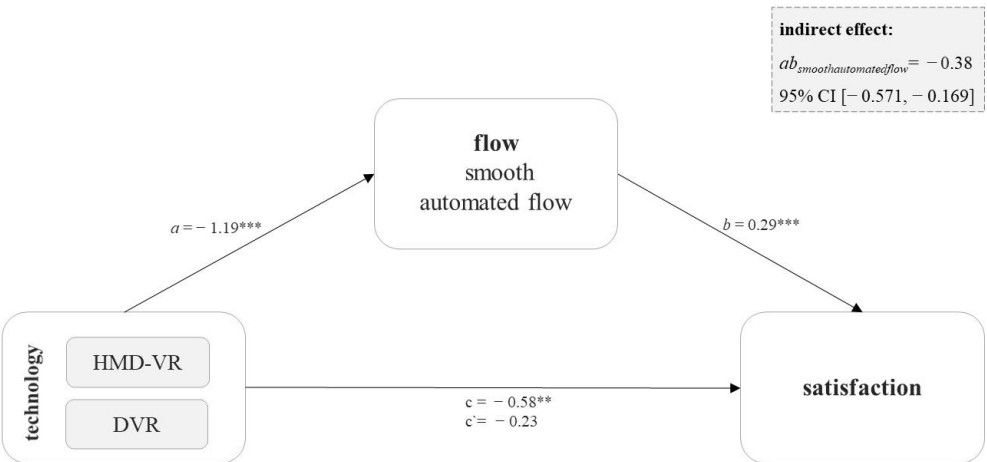

**Figure 7.** Presentation of a significant complete mediation—smooth automated flow—satisfaction. ** $p < 0.01$, *** $p < 0.001$.

Mediating effects through learning process variables were discovered a few times. Consequently, the experience of flow and presence within the VR environment can significantly explain the cause–effect relationships between VR technology and parameters of learning, even if only the effects for evaluative indicators could be found.

Nonetheless, subhypothesis a can be supported. Students experienced more physical presence when using an HMD than using a desktop, meaning that the first group mentioned tended to feel more spatially present in the virtual hiding place. HMDs largely block out external stimuli from reality, whereas on the desktop the real environment (e.g., the classroom) is still present. This is consistent with a study by Makransky and colleagues [64]. In addition, in this study, experiencing presence comes along with higher satisfaction and higher probability to recommend. Again, no mediation effects could be uncovered for other learning indicators. Contrary, in their study, Makransky et al. found negative effects of increased presence on learning.

Moreover, subhypothesis b can be accepted. The cause-and-effect relationships are identical to those reported for presence. Significantly more flow was experienced in the HMD-VR conditions than in the DVR conditions. When more flow was experienced, the application was rated significantly better.

It can be concluded that some learning process variables can significantly explain the causal relationships between the VR technology used and indicators of learning. These mediation effects are particularly relevant because presumed significant direct effects of technology on indicators of learning lose their significance when latent learning processes are also considered. Studies that fail to control for such learning processes occurring in VR misinterpret significant results between two variables as causal, when in fact other variables, namely latent learning processes, are responsible for or can systematically explain this relationship. Therefore, the results of the current study should encourage the inclusion of learning process variables such as flow and presence to avoid confounding.

## 6. Discussion

Regarding the overall relevance of the VR application *Anne Frank VR House* to support cognitive and affective learning, the following can be concluded: Across all experimental conditions, there is an increase in knowledge and perspective-taking into Anne Frank's situation. Considering the evaluation of the application, high means and low dispersion are detected resulting in a high satisfaction and a high tendency to recommend. Thus, the *Anne Frank VR House* seems to be conducive to middle school students, no matter which VR technology is used to transport the learning content.

Nevertheless, a significant effect for the variable VR technology was uncovered averaged across all learning indicators. However, a superiority of HMD-based VR was uncovered only for two evaluative indicators, and for one affective learning indicator, even if not statistically significant. The results for the cognitive indicators were ambiguous.

The instructional method of exposition seems to be beneficial for knowledge gain: Students in the exposition group have an advantage over students in the exploration group with respect to cognitive learning indicators.

With respect to prior (technical) knowledge, the majority of statistical analyses could not find any moderating effects of these control factors.

The feeling of presence and flow in VR was also rated as high, particularly in the HMD-VR conditions. Considering these factors as mediating variables, they can systematically explain the effects of certain VR technologies on learning. The results of the mediation analyses revealed that in several cases, not the technology itself, but the learning processes triggered by a certain technology, affect learning. This interesting finding still needs to be statistically corroborated, but it highlights the need of integrating learning process variables into future research designs.

## 7. Conclusions

In the present study, the *Anne Frank VR* House was empirically examined. Across the different experimental conditions, it was discovered that the application can address cognitive and affective learning goals and is evaluated very positively by the students. The question of which VR technology (HMD-VR vs. DVR) is more suitable for presenting the VR content cannot be fully answered. Unsurprisingly, significantly better ratings by students

for HMD-based VR environments compared to desktop-based VR environments were detected. For other cognitive and affective measures of learning, significant differences were found less frequently. For the cognitive learning indicators, the results were inconclusive or in favor of DVR. For the affective learning indicators, only numerically but not statistically significant advantages for HMD-based VR were apparent. Thus, a choice for or against a certain VR technology should always be a decision made depending on the teaching and learning goals. Another question addresses instructional methods accompanying the use of VR learning environments. In the current study, there were just some differences found between the methods of exposition and exploration. Only for the declarative knowledge acquisition slight advantages were shown in the exposition conditions.

More meaningful than the main effects of the VR technology and the instructional method were the moderation and mediation effects investigated in this study. Moderating effects due to prior (technical) knowledge were not found or were only found sporadically. Mediation effects, on the other hand, which can significantly explain the effect of VR technology on parameters of learning, were uncovered in several cases. The experience of presence and flow within VR could sufficiently explain the effects of technology on some learning indicators. The direct effect disappeared by adding a mediator and only the indirect effect mediated by the mediator proved to be statistically significant. At this point, it becomes evident why empirical studies ought to control for latent process variables. Otherwise, statistically significant direct effects might be falsely interpreted as significant or systematically overestimated, when in fact an underlying process is responsible for the relationship. Unfortunately, in this study, only mediation effects were found for the evaluative learning indicators.

Hence, the present study was able to exemplify that the investigation of complex interaction structures of VR technologies, instructional methods, learning process variables, and contextual factors can make a large contribution to the understanding of learning in VR environments. The study provides initial evidence for the underlying theoretical models [38,43,77]. However, many of the constructs listed in these models have remained untouched. The models thus continue to offer potential for further empirical work. In future research, it would be recommended to use other VR learning environments, other learning indicators, other instructional methods, for example, based on further principles of the Cognitive Theory of Multimedia Learning [81], and other learning process variables to develop and implement experimental designs.

**Funding:** The research received no external funding.

**Institutional Review Board Statement:** Ethical review and approval were waived for this study because all subjects participated in the study voluntarily and without further constraints.

**Informed Consent Statement:** Informed consent was obtained from all subjects involved in the study.

**Data Availability Statement:** Data Availability Statement: Data can be downloaded at: https://uni-duisburg-essen.sciebo.de/s/Eagf0g6jdSlUsry accessed on 12 February 2023. Further study materials can be downloaded at: https://uni-duisburg-essen.sciebo.de/s/fiIAIkkl1KTcutO accessed on 12 February 2023.

**Conflicts of Interest:** The author declares no conflict of interest.

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
