# Peer review of "Learning about Victims of Holocaust in Virtual Reality: The Main, Mediating and Moderating Effects of Technology, Instructional Method, Flow, Presence, and Prior Knowledge"

_mti, doi:10.3390/mti7030028_

Round 1
Reviewer 1 Report
This paper described 2x2 exploratory study that compares two technological approaches (DVR vs. HDM-VR), and two instructional methods (exposition vs. exploration). The study is deeply quantitative, exploring various variables and mediating effects that can be relevant to the design of educational technologies.
The main issue with the current paper draft is the complexity of the study, which makes it difficult to follow the main narrative. For example, Section 4.4 includes many statistical analyses; it would be helpful to have a discussion section (before the Conclusion) that highlights the key takeaways and describes design recommendations for the designers of educational technologies.
Additionally, some details of the statistical methods may need to be revised.
* The authors need to clarify the type of MANOVA used in the analysis. It sounds like they collected repeated measures, so I would have expected to see a MANOVA with repeated measures, maybe a doubly multivariate MANOVA. Rather, section 4.2 seems to suggest that “the pre value was subtracted from the post value.”
* Page 17: “After the mediator flow has been integrated into the analysis, the correlation between technology and satisfaction loses its statistical significance. Consequently, there is no direct link between technology and satisfaction” I am not sure I follow this point. Why does the lack of statistical significance imply “no direct link”? I would assume that the data simply cannot prove a direct link.
Other minor issues with this paper.
* The last sentence in the abstract is cryptic: “ Hence, the results pointed out that media comparison studies are short-sighted”
* Introduction: “more complex constellations of variables relevant to learning” Can you provide some examples?
* It would be interesting to add a teaser picture in the introduction to quickly introduce the prototypes (which is not introduced until page 8)
* Subsection 2.5 is key to understanding the experimental design, but it is buried in the related work section. I’d suggest making it a separate section (e.g., problem statement).
* A study that might be relevant for perspective-taking in learning environments:
Jessica Roberts, Leilah Lyons, Francesco Cafaro, and Rebecca Eydt. 2014. Interpreting data from within: supporting human data interaction in museum exhibits through perspective taking. In Proceedings of the 2014 conference on Interaction design and children (IDC '14). Association for Computing Machinery, New York, NY, USA, 7–16. https://doi.org/10.1145/2593968.2593974
Author Response
Thank you so much for your comments. Find my answers and solutions in the documents attached. If there are any questions, do not hesitate to contact me. Best regards!

Reviewer 2 Report
This paper presents one of the more thorough comparisons of HMD and desktop-based VR I've seen.
The study is well designed, although there are quite a few hypotheses being examined at once.
The experimental setup is described clearly, with useful details about the technology, tasks, users and so on.
Results are presented in some detail. Some of the presentation of statistics is a little unclear. For example, in section 4.2 you are listing MANOVA results with p values described as p > .00. Surely this is not correct?
It would also be useful to see some of the descriptive variables presented visually as well as in the table form.
My only real issue with this work would be the conclusion section. You present a lot of data in the results section, but there is minimal analysis and discussion of what this means. The conclusion is extremely short. I feel the paper would benefit from more discussion of what the results say for each hypothesis and what this means for the use of VR in such educational scenarios.
Author Response

(The authors gave the same response as above.)
